# Naturalizing a Programming Language via Interactive Learning

## Abstract

Our goal is to create a convenient language interface for performing well-specified but complex actions such as analyzing data, manipulating text, and querying databases. However, existing natural language interfaces for such tasks are quite primitive compared to the power one wields with a programming language. To bridge this gap, we seed the system with a core programming language and allow users to "naturalize" the core language incrementally by defining alternative syntax and increasingly complex concepts in terms of compositions of simpler ones. In a voxel world, we show that a community of users can simultaneously teach one system a diverse language and use it to build 240 complex voxel structures. Over the course of three days, these builders went from using only the core language to using the full naturalized language in 74.7% of the last 10K utterances.

## 1 Introduction

In tasks such as analyzing and plotting data, querying databases, manipulating texts, or controlling the Internet of Things, people need computers to perform well-specified but complex actions. To accomplish this, one route is to use a programming language, but this is inaccessible to most and can be tedious even for programmers. Another route is to convert natural language into a formal language, which has been the subject of work in semantic parsing (Zettlemoyer and Collins, 2005; Artzi and Zettlemoyer, 2013; Kushman and Barzilay, 2013; Quirk et al., 2015; Pasupat and Liang, 2015). However, the expressivity of semantic

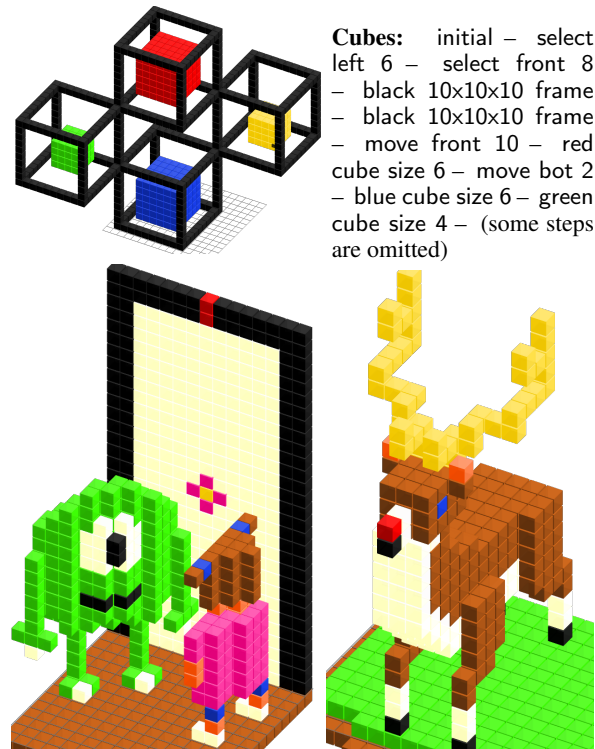

**Cubes:** initial – select left 6 – select front 8 – black 10x10x10 frame – black 10x10x10 frame – move front 10 – red cube size 6 – move bot 2 – blue cube size 6 – green cube size 4 – (some steps are omitted)

**Monsters, Inc:** initial – move forward – add green monster – go down 8 – go right and front – add brown floor – add girl – go back and down – add door – add black column 30 – go up 9 – finish door – (some steps for moving are omitted)

**Deer:** initial – bird's eye view – deer head; up; left 2; back 2; { left antler }; right 2; {right antler} – down 4; front 2; left 3; deer body; down 6; {deer leg front}; back 7; {deer leg back}; left 4; {deer leg back}; front 7; {deer leg front} – (last step censored)

Figure 1: Users interactively naturalize the language in Voxelurn (http://anonymous.url). Three of the 230 structures are shown.

parsers is still quite primitive compared to the power one wields with a programming language.

In this paper, we propose bridging this gap with a new interactive language learning process which we call *naturalization*. We seed a system with a core programming language, always available to the user. As users instruct the system to per-

form actions, they augment the language by *defining* new constructions — e.g., the user can explicitly tell the computer that 'X' means 'Y'. Through this process, the user gradually teaches the system to understand the language that they *want to use*, rather than the core language that users are forced to use initially. This process accommodates both the user preference and the computer action space, which tends to result in a combination of English, short forms, and precise commands suitable for the computer.

Compared to interactive language learning with weak denotational supervision (Wang et al., 2016), our use of definitions is critical for increasing the scope of what can be learned. Definitions equate a novel utterance to a sequence of utterances the system already understands. For example, 'go left 6 and front' might be defined as 'repeat 6 [go left]; go front', which eventually can be traced back to the expression 'repeat 6 [select left of this]; select front of this' in the core language. Unlike in programming, the user uses concrete values rather than variables, which makes creating our definitions more accessible. The onus is now on the system to produce the correct generalization, so we propose a grammar induction algorithm tailored for the definitions setting. Compared to standard machine learning, say from demonstrations, definitions provide a much more powerful learning signal: the system is told directly that 'a 3 by 4 red square' is '3 red columns of height 4', and does not have to induce this from observing many structures of different sizes.

We implemented a system called Voxelurn, which is a command interface for a voxel world initially equipped with a programming language supporting conditionals, loops, and variable scoping. We recruited 70 users from Amazon Mechanical Turk to build 300 voxel structures using our system. All users teach the system at once, so that what is learned immediately generalizes across users. It is thus a *community* of users that evolves the language to becomes more efficient over time. We show that the community defined many new utterances—shorter and alternative expressions, different syntax, more expressive operations, and also complex concepts such as 'green monster, yellow plate 3 x 3'. As the system learns, the expressiveness of the naturalized language grows, and users increasingly prefer to use the naturalized language over the core language

where 74.7% of of the last 10k accepted utterances are in the naturalized language.

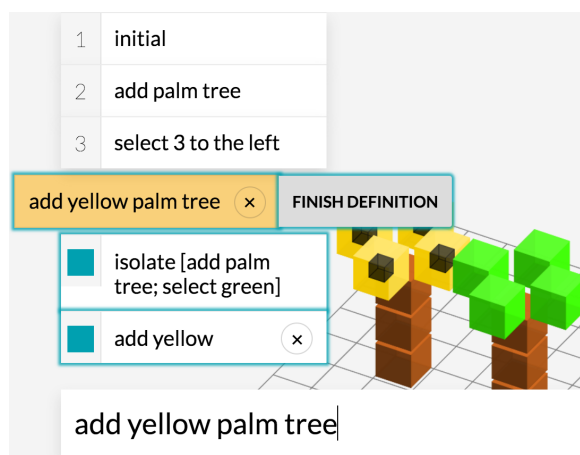

Figure 2: Interface used by users to perform actions and create definitions.

## 2 Voxelurn

**World.** A world state in Voxelurn contains a set of voxels, where each voxel has relations 'row', 'col'(column), 'height', and 'color'. There are two domain-specific actions, 'add' and 'move', one domain-specific relation 'direction'. In addition, the state contains a selection, which specifies a set of positions that subsequent actions are performed with respect to. Generally, we view the world as a set of similar objects equiped with some relations — events on a calendar, cells of a spreadsheet, and lines of text are other examples.

**Core language.** The core language for manipulating states in Voxelurn is a programming language that the system is born understanding (see Table 1 for an overview). We protect the core language from being redefined so it is always precise and usable.[1]

The language has expressive control primitives—'if', 'foreach', 'repeat', etc. Actions operate on sets, which are represented using basic lambda DCS expressions (Liang, 2013). Besides standard set operations like union, intersection and complement, Lambda DCS leverages the tree dependency structure common in natural language: for the relation 'color', 'has color red' refers to the set of voxels that has color red, and its reverse 'color of this' refers to the set of colors of the current selected voxels. Tree-structured

---

[1] We found that not doing so resulted in ambiguities that propagated uncontrollably, once 'red' means all colors.

| Rule(s) | Example(s) | Description |
|---|---|---|
| $A \rightarrow A; A$ | select left; add red | perform actions sequentially |
| $A \rightarrow$ repeat $N$ $A$ | repeat 3-1 add red top | repeat action $N$ times |
| $A \rightarrow$ if $S$ $A$ | if has color red [select origin] | action if $S$ is non-empty |
| $A \rightarrow$ while $S$ $A$ | while not has color red [select left of this] | action while $S$ is non-empty |
| $A \rightarrow$ foreach $S$ $A$ | foreach this [remove has row row of this] | action for each item in $S$ |
| $A \rightarrow [A]$ | [select left or right; add red; add red top] | group actions for precedence |
| $A \rightarrow \{A\}$ | {select left; add red} | scope only selection |
| $A \rightarrow$ isolate $A$ | isolate [add red top; select has color red] | scope voxels and selection |
| $A \rightarrow$ select $S$ | select all and not origin | set the selection |
| $A \rightarrow$ remove $S$ | remove has color red | remove voxels |
| $A \rightarrow$ update $R$ $S$ | update color [color of left of this] | change property of selection |
| $S$ | this | current selection |
| $S$ | all \| none \| origin | all voxels, empty set, $(0,0)$ |
| $R$ of $S$ \| has $R$ $S$ | has color red or yellow \| has row [col of this] | lambda DCS joins |
| not $S$ \| $S$ and $S$ \| $S$ or $S$ | this or left and not has color red | set operations |
| $N$ \| $N{+}N$ \| $N{-}N$ | 1,...,10 \| 1+2 \| row of this + 1 | numbers and arithmetic |
| argmax $R$ $S$ \| argmin $R$ $S$ | argmax col has color red | superlatives |
| $R$ | color \| row \| col \| height \| top \| left \| $\cdots$ | voxel relations |
| $C$ | red \| orange \| green \| blue \| black \| $\cdots$ | color values |
| $D$ | top \| bot \| front \| back \| left \| right | direction values |
| $S \rightarrow$ very $D$ of $S$ | very top of very bot of has color green | syntax sugar for argmax |
| $A \rightarrow$ add $C$ $[D]$ \| move $D$ | add red \| add yellow bot \| move left | add voxel, move selection |

Table 1: Grammar of the core programming language, which includes actions ($A$), relations ($R$), and sets of values ($S$). The grammar rules are grouped into four categories. From top to bottom: domain-general action compositions, actions using sets, lambda DCS expressions for sets, and domain-specific relations and actions.

joins can be chained without using any variables, so that 'has color [yellow or color of has row 1]' refers to all voxels with color yellow or those colors found in voxels in the first row.

In addition to expressivity, the core language *interpolates* well with natural language. We avoid explicit variables by using a *selection*, which serves as the default argument for most actions.[2] For example, 'select has color red; add yellow top; remove' adds yellow on top of red voxels and then removes the red voxels.

To enable the building of more complex structures in a more modular way, we introduce a notion of *scoping*. Suppose one is operating on one of the palm trees in Figure 2. The user might want to use 'select all' to select only the voxels in that tree rather than all of the voxels in the scene. In general, an action $A$ can be viewed as taking a set of voxels $v$ and a selection $s$, and producing an updated set of voxels $v'$ and a modified selection $s'$.

There are two constructs that alter the flow: First, '$\{A\}$' takes $(v, s)$ and returns $(v', s)$, thus restoring the selection. This allows one $A$ to use the selection as a temporary variable without affecting the rest of the program. Second, 'isolate $[A]$' takes $(v, s)$, calls $A$ with $(s, s)$ (restricting the set of voxels to just the selection) and returns $(v'', s)$, where $v''$ consists of voxels in $v'$ and voxels in $v$ that occupy empty locations in $v'$. This allows $A$ to only focus on the selection (e.g., one of the palm trees). Although scoping can be explicitly controlled via '[]', 'isolate', and '$\{\}$', it is an unnatural concept for non-programmers. Therefore, we let each expression generate three possible scoping interpretations, and let the model learn which one is intended based on the context. This is an example where we have preemptively started to naturalize the core language to reduce the burden on users.

---

[2]The selection is like the turtle in LOGO, but can be a set.

## 3 Learning interactively from definitions

The goal of the user is to build a structure in Voxelurn. In Wang et al. (2016), the user provided interactive supervision to the system by selecting from a list of candidates proposed by the system. This is practical when there are less than 50 candidates, but is completely infeasible for a complex action space such as Voxelurn. Roughly, 10 possible colors over the $3 \times 3 \times 4$ box containing the palm tree in Figure 2 yields $10^{36}$ distinct denotations, and many more programs. Obtaining the the structures in Figure 1 in this manner seems even more far-fetched.

To scale to these more complex structures, we allow the user to provide *definitions* in addition to the usual interactive supervision of Wang et al. (2016). Each definition consists of a *head* utterance and a *body*, which is a sequence of utterances that the system understands. There are two main use cases of definitions: The first is paraphrasing, which helps naturalizes the core language (e.g., defining 'add brown top 3 times' as 'repeat 3 add brown top'). The second is for the user to build up more complex concepts; in Figure 2, 'add yellow palm tree' is defined as a sequence of steps for building the palm tree. Once the system understands the *head*, it can be used in the body of other definitions; see Figure 3 for the full definition tree of 'add palm tree', after which one can build an entire forest of them.

---

**def:** add palm tree
 **def:** brown trunk height 3
 **def:** add brown top 3 times
 repeat 3 [add brown top]
 **def:** go to top of tree
 select very top of has color brown
 **def:** add leaves here
 **def:** select all sides
 select left or right or front or back
 add green

**Figure 3:** Defining 'add palm tree', tracing back to the core language (utterances without **def:**).

---

The interactive definition process is described in Figure 4. When the user types an utterance $x$, the system generates candidate parses. If the user selects one, then we simply execute the resulting program. If there are no parses or the user rejects all of the existing ones, the user is asked to provide the definition body for $x$. Any utterances

---

**begin** execute $x$:
 **if** *x does not parse* **then** define $x$;
 **if** *user rejects all parses* **then** define $x$;
 execute user choice
**begin** define $x$:
 **repeat** starting with $X \leftarrow []$
 user enters $x'$;
 **if** *x' does not parse* **then** define $x'$;
 **if** *user rejects all $x'$* **then** define $x'$;
 $X \leftarrow [X; x']$;
 **until** *user accepts $X$ as the def'n of $x$*;

**Figure 4:** when the user enters an utterance, the system tries to parse and execute it, or requests that the user define it.

---

in the body not yet understood can be defined recursively. Alternatively, the user can carry out a sequence of steps and mark these as the body of some head specified post-hoc.

When constructing the definition body, users can type utterances with multiple parses; e.g., 'move forward' could either modify the selection ('select front') or move the voxel ('move front'). Rather than propagating this uncertainty forward, we force the user to commit to an interpretation. This demonstrates the utility of interactivity in stopping a combinatorial explosion of interpretations.

## 4 Model and Learning

Let us turn to how the system learns and predicts. This section contains prerequisites before we can describe definitions and grammar induction in the next section.

**Semantic parsing.** Our system is based on a semantic parser that maps utterances $x$ to programs $z$, which can be executed on the current state $s$ (set of voxels and selection) to produce the next state $s' = [\![z]\!]_s$. Our system is implemented in SEMPRE (Berant et al., 2013); see Liang (2016) for a gentle exposition.

A *derivation* $d$ represents the process by which the utterance $x$ turns into a program $z = \text{prog}(d)$. More precisely, $d$ is a tree where each node contains a list of children derivations $[d_1, \ldots, d_n]$, the corresponding span of the utterance $(\text{start}(d), \text{end}(d))$, and grammar rule $\text{rule}(d)$, and the grammar category $\text{cat}(d)$.

Following Zettlemoyer and Collins (2005), we define a log-linear model over derivations $d$ given

| Feature | Description |
|---|---|
| Rule.ID | ID of the rule |
| Rule.Type | core?, used?, used by others? |
| Social.Author | ID of author |
| Social.Friends | (ID of author, ID of user) |
| Social.Self | rule is authored by user? |
| Span | (left/right token(s), category) |
| Scope | type of scoping for each user |

Table 2: Summary of features.

an utterance $x$ produced by the user $u$:

$$p_\theta(d \mid x, u) \propto \exp(\theta^\mathsf{T}\phi(d, x, u)), \qquad (1)$$

where $\phi(d, x, u) \in \mathbb{R}^p$ is a feature vector and $\theta \in \mathbb{R}^p$ is a parameter vector. The user $u$ does not appear in previous work on semantic parsing, but we use it to personalize the semantic parser trained on the community.

We use a standard chart parser to construct a chart. For each chart cell, indexed by the start and end indices of a span, we construct a list of partial derivations recursively by selecting children derivations from subspans and applying a grammar rule. The resulting derivations are sorted by model score and only the top $K$ are kept. We use $\text{chart}(x)$ to denote the set of all partial derivations across all chart cells. The set of grammar rules starts with the set of rules for the core language (Table 1), but grows via grammar induction when users add definitions (Section 5).

**Features.** Derivations are scored using a weighted combination of features. There are three types of features, summarized in Table 2.

**Rule features** fire on each rule used to construct a derivation. ID features fire on specific rules (by ID). Type features track whether a rule is part of the core language or user-induced, whether it has been used again after it was defined, if it was used by someone other than its author, and if the user and the author are the same. ($5 + \#$rules features)

**Social features** fire on properties of rules that capture the unique linguistic styles of different users and their interaction with one other. Author features capture the fact that some users provide better, and more generalizable definitions that tend to be used. Friends features are cross products of author ID and user ID, which captures whether rules from a particular author is systematically preferred or not by the current user, due stylistic similarities or differences. ($\#$users $+ \#$users $\times \#$users features)

**Span features** are conjunctions of the category with 2 tokens outside the span, and 1 token inside the span. These capture a weak form of context-dependence that are generally helpful. ($\approx V^4 \times \#$cats features for a vocabulary of size $V$)

**Scoping features** track how the community, as well as individual users prefer each of 3 scoping choices (none, selection only, and voxels+selection), as described in Section 2. There are 3 global indicators, and 3 indicators for each user firing every time a particular scoping choice is made. ($3 + 3 \times \#$users features)

**Parameter estimation.** When the user types an utterance, the system generates a list of candidate next states. When the user chooses a particular next state $s'$ from this list, the system performs a online AdaGrad (Duchi et al., 2010) update on the paratmers $\theta$ according to the gradient of the following loss function:

$$-\log \sum_{d : [\![\text{prog}(d)]\!]_s = s'} p_\theta(d \mid x, u) + \lambda ||\theta||_1,$$

which attempts to increase the model probability on derivations whose programs produce the next state $s'$.

## 5 Grammar Induction

Recall that the main form of supervision is via user definitions, which allows naturalization of the core language and creation of user-defined concepts. In this section, we show how to turn these definitions into new grammar rules that can be used by the system to parse new utterances.

Previous work on grammar induction for semantic parsing is given pairs of utterance-program pairs $(x, z)$. Both GENLEX (Zettlemoyer and Collins, 2005) and higher-order unification (Kwiatkowski et al., 2010) algorithms over-generate rules that liberally associate parts of $x$ with parts of $z$. Though unpromsing rules are immediately pruned, many spurious rules are undoubtedly still kept. In the interactive setting, we must keep the number of candidates small to avoid a bad user experience, our precision bar for new rules is much higher.

Fortunately, the structure of definitions makes the grammar induction task easier. Rather than being given a utterance-program $(x, z)$ pair, we are given a definition, which consists of an utterance

$x$ (head) along with the body $X = [x_1, \ldots, x_n]$, which is a sequence of utterances. The body $X$ is fully parsed into a derivation $d$, while the $x$ is likely only partially parsed into a set of partial derivations $\text{chart}(x)$.

At a high-level, we find *matches*—partial derivations $\text{chart}(x)$ of the head $x$ that also occur in the full derivation $d$ of the body $X$. A grammar rule is produced by substituting any set of non-overlapping matches by their categories. As an example, suppose the user defines

'add red top times 3' as 'repeat 3 [add red top]'.

Then we would be able to induce the following two grammar rules:

$$A \to \text{add } C \ D \text{ times } N \ :$$
$$\lambda CDN.\text{repeat } N \text{ [add } C \ D]$$
$$A \to A \text{ times } N :$$
$$\lambda AN.\text{repeat } N \ [A]$$

The first rule substitutes primitive values ('red', 'top', and '3') with their respective pre-terminal categories ($C$, $D$, $N$). The second rule contains compositional categories like actions ($A$), which requires some care. One might expect that greedily substituting the largest matches or the match that covers the largest portion of the body would work, but the following example that this is not the case:

$$\underbrace{\overbrace{\text{add red left}}^{A_1} \text{ and here}}_{A_2} = \underbrace{\overbrace{\text{add red left}}^{A_1}}_{A_2}; \overbrace{\text{add red}}^{A_1} \tag{2}$$

Here, both the highest coverage substitution ($A_1$: 'add red', which covers 4 tokens of the body), and the largest substitution available ($A_1$: 'add red left') would generalize incorrectly. The correct grammar rule only substitutes the primitive values ('red', 'left').

### 5.1 Highest scoring abstractions

We now propose a grammar induction procedure that optimizes a more global objective and uses the learned semantic parsing model to choose substitutions. To make things more formal, let $M$ be the set of partial derivations in the head whose programs also appear in the derivation $d$ of the body

$X$:

$$M \stackrel{\text{def}}{=} \{d' \in \text{chart}(x) :$$
$$\exists d'' \in \text{desc}(d) \land \text{prog}(d') = \text{prog}(d'')\},$$

where $\text{desc}(d)$ are partial derivations that are the descendants of $d$. Our goal is to find a *packing* $P \subseteq M$, which is a set of derivations corresponding to non-overlapping spans. We say that a packing $P$ is maximal if there is no other derivations may be added without creating an overlap.

Letting $\text{packing}(M)$ be the set of all maximal packings, we can now frame our problem as finding the maximal packing that has the highest score under our current semantic parsing model:

$$P_l^* = \operatorname*{argmax}_{P \in \text{packing}(M);} \sum_{d \in P} \text{score}(d).$$

Finding the highest scoring packing can be done using dynamic programming on $P_i^*$ for $i = 1, \ldots, l$, where $l$ is the length of $x$. To obtain this dynamic program, let $D_i$ be the highest scoring maximal packing containing a derivation ending exactly at position $i$,

$$D_i = \{d_i\} \cup P^*_{\text{start}(d)}, \tag{3}$$
$$d_i = \operatorname*{argmax}_{d \in M; \text{end}(d)=i} \text{score}(d \cup P^*_{\text{start}(d)}). \tag{4}$$

Then the maximal packing of up to $i$ can be defined recursively as

$$P_i^* = \operatorname*{argmax}_{D \in \{D_{s+1}, D_{s+2}, \ldots, D_i\}} \text{score}(D) \tag{5}$$
$$s(i) = \max_{d:\text{end}(d) \leq i} \text{start}(d), \tag{6}$$

where $s = s(i)$ is the largest index such that $D_{s(i)}$ is no longer maximal for the span $(1, i)$ (i.e there is a $d' \in M$ on the span $\text{start}(d) >= s(i) \land \text{end}(d) \leq i$.

Once we have a packing $P^* = P_l^*$, we can go through $d' \in P^*$ in order of $\text{start}(d')$, as in Algorithm 1.

Algorithm 1 generates one high precision rule per packing per definition. In addition to the highest scoring packing, we also use the "simple packing", which includes only primitive values (in Voxelurn, these are colors, numbers, and directions). We also attempt to generalize compositional categories like sets and actions. Here correctness is not guaranteed—however a rule that often generalizes incorrectly will be down-weighted,

```
Input   : x, d, P*
Output: rule
r ← x;
f ← d;
for d' ∈ P* do
    r ← r[span(d') ← cat(d')]
    f ← λ cat(d').f[d' ← cat(d')]
return rule cat(d)→ r : f
```

**Algorithm 1:** Extract a rule $r$ from a derivation $d$ and a packing $P^*$. Here, $f[s \leftarrow t]$ means substitute $s$ with $t$ in $f$, with the usual care taken about names in variable bindings.

along with the score of its packings, relative to the score of the simple packing. As a result, a different rule might be induced, even with the same inputs $x$ and $d$.

### 5.2 Extending the chart via alignment

The previous procedure yields high precision rules, but fails to generalize enough. Suppose that 'move up' is defined as 'move top'. But since 'up' does not parse, it also does not match anything, whereas we would like to infer that 'up' means 'top'.

To handle this, we leverage a property of definitions that we haven't been using thus far: the utterances themselves. If we align the head and body, then we would intuitively expect aligned phrases to correspond to the same derivations. We can then transplant these derivations from $d$ to $\text{chart}(x)$ to create some new matches. This is more constrained than the usual alignment problem (e.g., in machine translation) since we only need to consider spans of $X$ which corresponds to derivations in $\text{desc}(d)$.

```
Input   : x, X, d
for d' ∈ desc(d), x' ∈ spans(x) do
    if aligned(x', d', (x, X)) then
        d_t ← d';
        start(d_t) ← start(x');
        end(d_t) ← end(x');
        chart(x) ← chart(x) ∪ d_t
    end
end
```

**Algorithm 2:** Extending the chart by alignment: If $d'$ is aligned with $x'$ based on the utterance, then we pretend that $x'$ should also parse to $d'$, and $d'$ is transplanted to $\text{chart}(x)$ as if it came from $x'$.

Algorithm 2 provides the algorithm for extending the chart via alignments. The aligned function is implemented using the following two heuristics:

- **exclusion**: if all but 1 pair of short spans are matched, the unmatched pair is considered aligned.

- **projectivity**: if $d_1, d_2 \in \text{desc}(d) \cap \text{chart}(x)$, then $\text{ances}(d_1, d_2)$ is considered aligned to the corresponding span in $x$.

With the extended chart, we can run the algorithm from Section 5.1 to induce rules. The transplanted derivations (e.g., 'up') might now form new matches which allows the grammar induction for induce more generalizable rules. We only perform this extension when the body consists of one utterance, which tend to be paraphrases. Multi-utterance definitions tend to be about constructing new concepts, for which extension is not useful. Because low precision rules significantly degrades user experience in the interactive setting, we induce alignment based rules conservatively—only when 1 or 2 tokens were left unmatched.

## 6 Experiments

**Setup.** Our ultimate goal is to create a community of users who can build interesting structures in Voxelurn while naturalizing the core language. We created this community using Amazon Mechanical Turk (AMT) in two stages. First, we have *qualifier* tasks, in which an AMT worker was instructed to build an exact simple, fixed target that we provide . In addition, replicating a target ensures that the users are able to use basic constructs of the core language, which is the starting point of the naturalization process.

Next, we allowed the workers who qualified to enter the second *freebuilding* stage, in which they were asked to build any structure they wished in 30 minutes. This two stage process was designed to give users freedom while guarding against spam.

**Incentives.** To incentivize good structures, we setup a leaderboard which ranked structures based on recency and upvotes (Hacker News) Over the course of 3 days, we picked three prize categories to be released daily. The prize categories for each day were bridge, house, animal – tower, monster(s), flower(s) – ship(s), dancer(s), and castle.

Finally, to incentivize more definitions, we also track *citations*. When a rule is used in an accepted

| **Short forms** |
| --- |
| left, l, mov left, go left, <, sel left |
| br, blk, blu, brn, orangeright, left3 |
| add row brn left 5 := add row brown left 5 |
| **Syntactic** |
| go down and right := go down; go right |
| select orange := select has color orange |
| add red top 4 times := repeat 4 [add red top] |
| l white := go left and add white |
| mov up 2 := repeat 2 [select up] |
| go up 3 := go up 2; go up |
| **Higher level** |
| add black block width 2 length 2 height 3 := {repeat 3 [add black platform width 2. . . |
| flower petals := flower petal; back; flower petals |
| cube size 5, get into position start, |
| 5 x 5 open green square, brownbase |

Table 3: some definitions on the leaderboard.

utterance by another user, the rule (and its author) gets a citaion. We pay bonuses to top users according to their h-index.

**Statistics.** 70 workers were granted the qualification, and 42 workers participated in the final free building experiment, who built 230 structures. Users made 64,075 utterances, of these 36,589 are accepted (so the action is performed). There were 2,495 definitions resulting in 2,817 grammar rules, compared to less than 100 core rules.

**Is naturalization happening?** Yes! according to Figure 5, which plots the cummulative percentage of utterances that belong to core, induced, or none (which means we did not parse it). At the end, 58% of the utterances are parsed with induced rules. To rule out that these utterances are getting rejected more often, we consider only accepted utterances in the middle plot of Figure 5, which plots the percentage of induced rules among accepted utterances for the entire community, as well as for the 5 heaviest users. At the end, 64.3% of all accepted utterances are induced, which is 77.9% in the last 10,000 accepted utterances.

We see three interpretable modes of naturalization outlined in Table 3. For very common operations, like moving the selection, people found 'select left' too verbose and shorterned it. For syntactic variations, where many of the definition body are not in core. The definitions for high level concepts tend long and hierarchical. The bottom plot of Figure 5 shows that users are building higher level concepts, since programs become

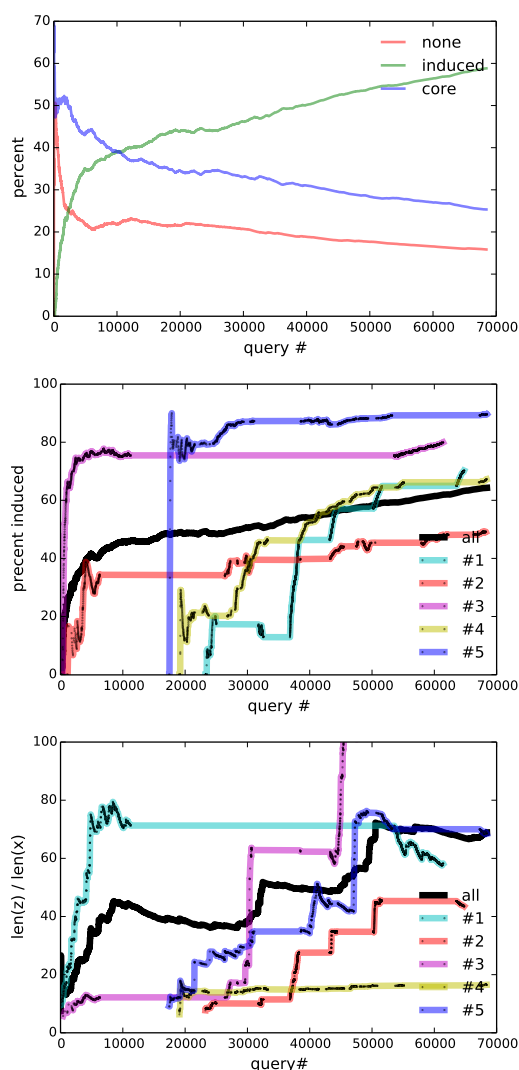

Figure 5: learning curves. **top:** percentage of all utterances belonging to each type. **mid:** percentage of accepted utterances belonging to induced. **bot:** expressiveness measured by the average length of the program per token of the utterance.

## 7 Discussion

We applied our methodology to Voxelurn, but it should apply to other settings: answering questions about semi-structured data (Pasupat and Liang, 2015), parsing to regular expressions (Kushman and Barzilay, 2013), or following robot instructions (Chen and Mooney, 2011; Tellex et al., 2011; Artzi and Zettlemoyer, 2013; Misra et al., 2015). In these applications, instead of learning from a training set, we could start with the programming language. User interactions can provide us with very strong, usable supervision through definitions. We hope that naturalization can lead to better language interface technologies that strikes a more usable balance between precision and naturalness.

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
