# Peer review of "Naturalizing a Programming Language via Interactive Learning"

_ACL 2017 — decision unknown_

[Official Review · Reviewer 1 · rating 4 · confidence 4]
soundness 3 · originality 4 · clarity 4 · impact 5 · substance 4 · appropriateness 5 · meaningful comparison 5 · presentation format Oral Presentation

Thanks for the response. I look forward to reading about the effect of
incentives and the ambiguity of the language in the domain.

Review before author response:
The paper proposes a way to build natural language interfaces by allowing a set
of users to define new concepts and syntax. It's an (non-trivial) extension of
S. I. Wang, P. Liang, and C. Manning. 2016. Learning language games through
interaction

Questions:
- What is the size of the vocabulary used 
- Is it possible to position this paper with respect to previous work on
inverse reinforcement learning and imitation learning ?

Strengths:
- The paper is well written
- It provides a compelling direction/solution to the problem of dealing with a
large set of possible programs while learning natural language interfaces. 

Weaknesses:
- The authors should discuss the effect of the incentives on the final
performance ? Were other alternatives considered ? 
- While the paper claims that the method can be extended to more practical
domains, it is not clear to me how straightforward it is going to be. How
sensitive is the method to the size of the vocabulary required in a domain ?
Would increased ambiguity in natural language create new problems ? These
questions are not discussed in the current experiments.
- A real-world application would definitely strengthen the paper even more.

[Official Review · Reviewer 2 · rating 4 · confidence 4]
soundness 3 · originality 4 · clarity 4 · impact 5 · substance 4 · appropriateness 5 · meaningful comparison 5 · presentation format Oral Presentation

- Strengths: This paper reports on an interesting project to enable people to
design their own language for interacting with a computer program, in place of
using a programming language. The specific construction that the authors focus
on is the ability for people to make definitions. Very nicely, they can make
recursive definitions to arrive at a very general way of giving a command. The
example showing how the user could generate definitions to create a palm tree
was motivating. The approach using learning of grammars to capture new cases
seems like a good one. 

- Weaknesses: This seems to be an extension of the ACL 2016 paper on a similar
topic. It would be helpful to be more explicit about what is new in this paper
over the old one. 

There was not much comparison with previous work: no related work section. 

The features for learning are interesting but it's not always clear how they
would come into play. For example, it would be good to see an example of how
the social features influenced the outcome. I did not otherwise see how people
work together to create a language. 

- General Discussion:

[Official Review · Reviewer 3 · rating 4 · confidence 3]
soundness 3 · originality 4 · clarity 5 · impact 5 · substance 4 · appropriateness 5 · meaningful comparison 5 · presentation format Oral Presentation

- Strengths:

The ideas and the task addressed in this paper are beautiful and original.
Combining indirect supervision (accepting the resulting parse) with direct
supervision (giving a definition) makes it a particularly powerful way of
interactively building a natural language interface to a programming language.
The proposed has a wide range of potential applications. 

- Weaknesses:

The paper has several typos and language errors and some text seems to be
missing from the end of section 6. It could benefit from careful proofreading
by a native English speaker. 

- General Discussion:

The paper presents a method for collaborative naturalization of a 'core'
programming language by a community of users through incremental expansion of
the syntax of the language. This expansion is performed interactively, whereby
a user just types a command in the naturalized language, and then either
selects through a list of candidate parses or provides a definition also in the
natural language. The users give intuitive definitions using literals instead
of variables (e.g. "select orange"), which makes this method applicable to
non-programmers. 
A grammar is induced incrementally which is used to provide the candidate
parses.

I have read the authors' response.